# The Novel Role of the NLRP3 Inflammasome in Mycotoxin-Induced Toxicological Mechanisms

**DOI:** 10.3390/vetsci11070291

**Published:** 2024-06-28

**Authors:** Chengshui Liao, Fengru Xu, Zuhua Yu, Ke Ding, Yanyan Jia

**Affiliations:** 1Laboratory of Functional Microbiology and Animal Health, College of Animal Science and Technology, Henan University of Science and Technology, Luoyang 471023, China; liaochengshui33@haust.edu.cn (C.L.); 220320181461@stu.haust.edu.cn (F.X.); 9902818@haust.edu.cn (Z.Y.); 9903166@haust.edu.cn (K.D.); 2Luoyang Key Laboratory of Live Carrier Biomaterial and Animal Disease Prevention and Control, Luoyang 471023, China; 3The Key Lab of Animal Disease and Public Health, Henan University of Science and Technology, Luoyang 471023, China

**Keywords:** mycotoxins, pyroptosis, NLRP3 inflammasome, activation, inhibitor

## Abstract

**Simple Summary:**

Mycotoxins pose a serious threat to human and animal health by causing acute poisoning and chronic effects. However, the toxicological mechanism of mycotoxins is complicated and unclear. Recent reports have revealed that activation of the nucleotide-binding, oligomerization domain (NOD)-like receptor (NLR) family pyrin domain containing 3 (NLRP3) inflammasome is linked with the tissue damage and inflammation induced by mycotoxin exposure. Through a comprehensive literature review, this study illuminates the dysregulated expression of NLRP3 inflammasome responses to mycotoxin exposure. This study not only advances our knowledge of the role of the NLRP3 inflammasome in mycotoxin exposure but also offers valuable insights for future studies of novel anti-inflammatory agents used in cases of mycotoxin exposure.

**Abstract:**

Mycotoxins are secondary metabolites produced by several fungi and moulds that exert toxicological effects on animals including immunotoxicity, genotoxicity, hepatotoxicity, teratogenicity, and neurotoxicity. However, the toxicological mechanisms of mycotoxins are complex and unclear. The nucleotide-binding oligomerization domain (NOD)-like receptor (NLR) family pyrin domain containing 3 (NLRP3) inflammasome is a multimeric cytosolic protein complex composed of the NLRP3 sensor, ASC adapter protein, and caspase-1 effector. Activation of the NLRP3 inflammasome plays a crucial role in innate immune defence and homeostatic maintenance. Recent studies have revealed that NLRP3 inflammasome activation is linked to tissue damage and inflammation induced by mycotoxin exposure. Thus, this review summarises the latest advancements in research on the roles of NLRP3 inflammasome activation in the pathogenesis of mycotoxin exposure. The effects of exposure to multiple mycotoxins, including deoxynivalenol, aflatoxin B1, zearalenone, T-2 toxin, ochratoxin A, and fumonisim B1, on pyroptosis-related factors and inflammation-related factors in vitro and in vivo and the pharmacological inhibition of specific and nonspecific NLRP3 inhibitors are summarized and examined. This comprehensive review contributes to a better understanding of the role of the NLRP3 inflammasome in toxicity induced by mycotoxin exposure and provides novel insights for pharmacologically targeting NLRP3 as a novel anti-inflammatory agent against mycotoxin exposure.

## 1. Introduction

Mycotoxins are toxic secondary metabolites produced by fungi and are widely occurring in crops. In past years, mycotoxin contamination has been considered the most hazardous public health problem in food and feed, attracting significant attention worldwide [1]. Approximately 500 mycotoxins from toxic fungal metabolites have been estimated to have the potential to contaminate food and feed. Common mycotoxins include aflatoxins (AFTs), fumonisins B, deoxynivalenol (DON), ochratoxin A (OTA), T-2 toxin, zearalenone (ZEA), *Alternaria* toxins, HT-2 toxin, citrinin, enniatins, ergot alkaloids, nivalenol, cyclopiazonic acid, and patulin [1]. These mycotoxins can induce significant harm to the kidneys, livers, and reproductive organs of animals by triggering immune responses, inflammation, and oxidative stress (OS), consequently leading to substantial economic losses within the livestock industry [2,3]. Mycotoxins have long been recognized as a source of human food-borne illness and may be associated with several chronic diseases. The specific mechanism of mycotoxin-induced toxicity is unclear; however, previous research has shown that it is closely related to DNA damage, cell-cycle arrest, OS, the inflammatory response, and apoptosis. 

In recent years, a number of investigators have reported that the toxic effects of mycotoxins are related to pyroptosis [4,5,6]. The term "pyroptosis" was first used in 2001 to describe a type of inflammatory cell death characterized by the appearance of pores in the plasma membrane, cell swelling and rupture, and leakage of cell contents [7]. Pyroptosis is a key component of the immune response and is widely associated with the occurrence and progression of infectious diseases, tumors, nervous system-related diseases, metabolic diseases, and chronic inflammation [8]. The mechanisms of pyroptosis can be divided into classic and nonclassic pathways [9,10]. In the classic pathway, the NOD-, LRR-family and pyrin domain-containing protein 1 (NLRP1), NLRP3, NLR-family and caspase activation recruitment domain-containing protein 4 (NLRC4), absent in melanoma 2 (AIM2), pyrin, or other inflammasomes are activated through several specific ligands. For instance, the NLRC4 inflammasome is activated by flagellar and rod proteins that make up the components of the type III secretory apparatus. The AIM2 and IFN-gamma-inducible protein-16 (IFI16) inflammasomes recognize double-stranded DNA via the hemopoietic IFN-inducible nuclear proteins (HIN)-200 domain. Subsequently, the inflammasome complex is assembled to activate caspase-1, and activated caspase-1 cleaves gasdermin D (GSDMD) to generate active N- and C-termini. GSDMD-N causes cell membrane perforation, leading to cell death. Moreover, activated caspase-1 induces the conversion of pro-IL-1β to IL-1β, which is then released into the extracellular space where it amplifies inflammation. The nonclassic pyroptosis pathway mainly depends on the function of caspase-4/5 in humans (murine caspase-11) [11]. The ligand lipopolysaccharide (LPS) of Gram-negative bacteria directly activates caspase-4/5/11 without cleaving the precursors of interleukin-1β (IL-1β) and IL-18 to cleave GSDMD, and GSDMD-N is subsequently transferred to the plasmalemma. Activation of caspase-4/5/11 actuates the pannexin-1 channel, thereby releasing ATP and opening the membrane channel P2X7 to induce cell membrane lysis and pyroptosis [12].

As a key factor of pyroptosis, the NLRP3 inflammasome has been associated with numerous diseases, such as inflammatory bowel disease, infectious disease, and cancer [13]. Moreover, the application potential of targeting the NLRP3 inflammasome is supported by observations regarding therapies for autoimmune diseases [14]. NLRP3 inflammasome-mediated pyroptosis is instrumental in the process of mycotoxin-induced pathological damage [4]. Elucidating the mechanisms of inflammasome induction caused by mycotoxin exposure is crucial for circumventing mycotoxin-mediated pathology. In this review, we summarize the evidence for mycotoxin-induced damage and stimulation of the NLRP3 inflammasome. Furthermore, we highlight the role of the NLRP3 inflammasome in mycotoxin-induced toxicity. Finally, the possibility of targeting the NLRP3 inflammasome in cases of mycotoxin exposure is discussed.

## 2. Mycotoxicosis

Most mycotoxins are chemically stable under harsh conditions. Warm and humid environments are usually conducive to the growth of molds, further contributing to pollution caused by mycotoxins. There are hundreds of known mycotoxins, among which the most common harmful mycotoxins are mostly produced by *Aspergillus* spp., *Fusarium* spp., and *Penicillium* spp., among others. Mycotoxin contamination may occur at any stage of crop growth, harvest, storage, or feed processing. As noted by Eskola et al., up to 60–80% of food crop samples worldwide are threatened by mycotoxin contamination [15]. Ingestion of these mycotoxins can rapidly cause disease symptoms and may be life-threatening in severe cases. Mycotoxins can also cause long-term harm to humans, leading to cancer and immunodeficiency diseases [16]. Therefore, mycotoxin contamination poses a serious challenge to food safety and animal husbandry development and is a danger to human and animal health [17].

Food safety has received increasing attention, and research into the prevention and control of mycotoxin contamination has gradually become a greater focus of academic attention. Scientists have explored in depth the toxicity mechanisms of mycotoxins, which exert toxicological effects on animals through immunotoxicity, genotoxicity, OS, hepatotoxicity, cytotoxicity, teratogenicity, and neurotoxicity [18]. In vitro, mycotoxins induce programmed cell death such as apoptosis, as well as irreversible cell damage [19]. In addition, mycotoxins prevent the formation of mitochondrial complex I in cells, hindering the normal function of mitochondria and leading to increased generation of reactive oxygen species [20]. Moreover, mycotoxins have potassium-specific ionophore activity, which allows potassium ions to flow into the mitochondrial matrix and cause mitochondrial swelling [21]. Additionally, mycotoxins can bind to the 60S subunit of eukaryotic ribosomes and inhibit the production of peptide transferases [22]. Researchers have also found that mycotoxins competitively inhibit the activities of metabolic enzymes, thereby inhibiting carbohydrate and lipid metabolism [23]. Mycotoxins are structurally similar to sex hormones, affecting hormone receptors and altering hormone levels [24]. In addition, mycotoxins have a great impact on deoxyribonucleic acid (DNA) and ribonucleic acid (RNA). Aflatoxin B1 (AFB1) can indirectly inhibit the activity of RNA polymerase, thereby interfering with RNA synthesis [25]. Moreover, AFB1 undergoes irreversible covalent interactions with DNA, leading to the formation of N7–guanine adducts [26]. More importantly, a few mycotoxins can change DNA methylation levels [27]. In vivo studies have shown that mycotoxins induce different degrees of inflammatory responses and toxicological effects, resulting in organ lesions and even death in pigs, rats, chickens, mice, and other animals [28]. The diversity of mycotoxins and the complexity of toxicity mechanisms indicate that the control of mycotoxins activity is very difficult. It is crucial to investigate the toxicity of mycotoxins and elucidate their mechanisms of action to improve food safety and promote the development of animal husbandry. Numerous studies have demonstrated that NLRP3 inflammasome-mediated pyroptosis plays a critical role in mycotoxin-induced toxicity [4,5,6].

## 3. Mycotoxins Activate the NLRP3 Inflammasome

In 2002, Martinon and colleagues first described a caspase-activating complex using the “inflammasome” [29]. Inflammasomes are large multi-molecular arrays with important roles in regulating innate immunity and inflammation. Pyroptosis is an inflammatory type of programmed cell death caused by the activation of inflammasomes. Unlike cells undergoing apoptosis and necrosis, pyroptotic cells continue to swell until the cell membranes burst, releasing cellular debris and producing a strong inflammatory response [30]. Recently, as a novel method of programmed cell death, pyroptosis has been shown to be associated with many diseases [31]. The occurrence of pyroptosis depends on a variety of inflammasomes, among which the NLRP3 inflammasomes are one of the best-studied types. The NLRP3 inflammasomes are composed of an NLRP3 sensor, an apoptosis-associated speck-like protein containing a C-terminal caspase recruitment domain (ASC) adaptor and a caspase-1 effector. Data increasingly indicate that NLRP3 inflammasomes are widely found in epithelial and immune cells and are important in host defence [32]. The three main signaling pathways involved in the activation of NLRP3 inflammasomes are known as canonical, non-canonical, and alternate activation [32]. Activation of the canonical and non-canonical NLRP3 inflammasomes requires two steps: priming and activation. However, the alternative activation pathway requires only one step.

In the priming step for canonical activation, PRRs (TLRs or NLRs) or cytosolic receptors (PAMPs or DAMPs) activate nuclear factor kappa-light-chain-enhancer of activated B cells (NF-κB), and post-translational modifications (ubiquitination or phosphorylation) of NLRP3 promote its expression and that of IL-1β [33]. In the activation step, numerous molecular or cellular events occur, including ionic flux (efflux of potassium and chloride ions, influx of sodium, and calcium ion mobilization), mitochondrial dysfunction, pro-cesses involving mitochondrial DNA (mtDNA), reactive oxygen species (ROS) [34], and mitochondrial ROS (mtROS), trans-Golgi disassembly, lysosomal disruption, and metabolic changes [32]. Subsequently, the assembly and activation of NLRP3 inflammasomes stimulates caspase-1 activity and triggers the release of the cytokines IL-1β and IL-18, together with the subsequent activation of GSDMD, inducing an inflammatory response and cell death by lysis [35]. However, non-canonical activation is closely related to caspase-4 and -5 in humans and caspase-11 in mice [36]. In this signaling pathway, extracellular LPS activates toll-like receptor 4 (TLR4) and inducible type-I interferon, along with the complementary C3-C3aR system, upregulating caspase-11 expression [32]. GSDMD cleavage by activated caspases-4, -5, and -11 results in caspase oligomerization, autoproteolysis, and pyroptosis [37,38,39]. Along with the 2-step activation model, a 1-step NLRP3 inflammasome activation has been observed. Alternative inflammasome activation occurs in monocytes from humans and pigs and macrophages from bone marrow of mice. This pathway does not involve ASC speck formation, pyroptosis induction, or K^+^ efflux [32]. In this pathway, the TLR ligand alone is sufficient to activate maturity and release of caspase-1 or IL-1β from human or porcine monocytes via the TRIF/RIPK1/FADD caspase-8 pathway [33]. This one-step rapid response might be helpful in sustaining blood sterility in the early stages of infection and reducing bacterial septicaemia.

In recent years, NLRP3 inflammasomes have been to be activated in vitro and in vivo in response to exposure to mycotoxins including DON, AFB1, ZEA, T-2 toxin, OTA, and fumonisin B1 (FB1) (Figure 1).

### 3.1. DON and the NLRP3 Inflammasome

DON (vomitoxin) is a secondary fungal metabolite primarily produced by *Fusarium* spp. (*F. graminearum* and *F. culmorum*) that is a group B trichothecene mycotoxin. DON contaminates approximately 60% of wheat, corn, barley, rice, oats, sorghum, and rye, worldwide. The percentage of animal feeds contaminated by DON was 42% in the Middle East and North Africa during the years from 2012 to 2020 [40] and less than 20% in animal feeds from Italy over 5 years (2018–2022) [41]. However, the detection rate of DON was 71–76% in 2023 in animal feeds from northern Spain [42] and 78.91% in 2024 in swine feed from South Korea [43]. The molecular formula of DON (1R,2R,3S,7R,9R,10R,12S)-3,10-dihydroxy-2-(hydroxymethyl)-1,5-dimethylspiro[8-oxatri-cyclo [7.2.1.02,7] dodec-5-ene-12,2′-oxirane]-4-one) is C_15_H_20_O_6_. DON is heat-stable, with no reduction in DON concentration after 30 min at 170 °C; however, structural instability may help in detoxification [44]. DON exposure often causes strong toxicity to the intestinal, dermal, neurological, and reproductive systems, as well as immunotoxicity resulting in anorexia, diarrhoea, vomiting, weight loss, growth delay, immune disorders, and even carcinogenesis in animals [45]. Inflammation, mitochondrial disruption, and apoptosis through ROS are hypothesized as the primary mechanisms of DON toxicity [44].

The intestines are the critical target organ of DON, which can cause gut impairment, inflammatory reactions, and dysfunction; therefore, intestinal porcine epithelial cells (IPEC-J2) were employed as models for validating DON-induced intestinal damage. DON (2 µg/mL) activated NLRP3 inflammasomes by regulating OS and induced the production of proinflammatory cytokines (IL-18 and IL-1β), resulting in pyroptosis in IPEC-J2 cells [46,47]. Stress from accumulated ROS was instrumental in DON-inducible NLRP3-mediated pyroptosis; thus, caveolin-1 is a therapeutic target for overcoming DON-induced enterotoxicity [48]. Interestingly, DON exposure can aggravate LPS-induced cellular inflammatory responses and increase NLRP3 and procaspase-1 in IPEC-J2 cells through NF-κB signaling and the autophagy-related protein LC3B [49]. Thus, DON exposure could induce apoptosis and inflammation through ROS accumulation, NF-κB activation, and apoptosis in IPEC-J2 cells. However, incubation with non-toxic doses of DON (2 μM for IPEC-J2 cells) did not change the levels of NLRP3 or procaspase-1 [50].

DON-mediated activation of NLRP3 inflammasomes has been studied in different cells. Song et al. reported that incubation with DON increased the expression of caspase-1 and GSDMD in donkey (*Equus asinus*) endometrial epithelial cells [51]. However, determination of NLRP3 expression was not performed in their study. The synthesis of pro-IL-1β and NLRP3 by the action of NF-κB are the primary signals of NLRP3 inflammasome activation. DON can upregulate IL-1β in murine BV2 microglia and promote the formation of ASC/NLRP3 inflammasomes by activating the NF-κB signaling pathway [52]. Another study indicated that lymphocytes from the spleens of carp (*Cyprinus carpio* L.) incubated with DON had higher levels of mtROS, disrupting the balance of mitochondrial homeostasis and activating the mtROS/NF-κB/NLRP3 axis, and inducing pyroptosis [53]. In human keratinocytes (HaCaT cells), DON exposure also significantly induced excess mtROS production, disrupted mitochondria, and increased OS, apoptosis, and pyroptosis by activating MAPK/NF-κB and NLRP3 signaling [54]. Moreover, DON induced OS and activated the Keap1/Nrf2 and TLR4/NF-κB signaling pathways that contribute to the production of inflammatory cytokines. The above data show that OS is a primary signal in the DON-induced activation of NLRP3. Deafness autosomal dominant 5 (DFNA5), also called gasdermin D (GSDME), belongs to the gasdermin family. A study by Roger et al. [55] revealed a novel mechanism by which GSDME activated caspase-3-mediated regulated pyroptosis. After exposure to 32 μM or 64 μM DON for 24 h, HepaRG cells exhibited typical pyroptotic characteristics, including the release of IL-1β and IL-6 and the activation of caspase-3 and GSDME [56]. In addition, at concentrations approximating those in bovine follicular fluids, DON activated NLRP3 inflammasomes in ovarian thecal cells. Because of the association between inflammation of the ovaries, ageing and infertility, the effect of DON on dairy cow fertility should be studied further [57].

In vivo, DON-induced NLRP3-dependent pyroptosis was detected in mice and piglets. Several studies have reported that chronic and subacute oral administration of DON induced liver inflammatory injury and intestinal damage in mice by activating caspase-3/GSDME-dependent pyroptosis [56,58]. DON exposure activated NLRP3 inflammasomes, increased OS, and induced the expression of pyroptosis-related factors (GSDMD, ASC, caspase-1 P20, and IL-1β) in the mouse jejunum [48]. DON treatment enhanced OS in mice and piglets infected with enterotoxic *Escherichia coli* (ETEC) and increased the activation of NLRP3 inflammasomes in the jejunum and the expression of NLRP3, caspase-1, and ASC [50]. In addition, DON exposure aggravated porcine epidemic diarrhea virus (PEDV)-induced immunosuppression by inhibiting TLR4/NF-κB/NLRP3 signaling in weaned piglets, thereby causing intestinal mechanical/immune barrier dysfunction [59]. Although DON exposure has been confirmed to activate the NLRP3 inflammasome and cause host inflammatory damage in vitro and in vivo, the comprehensive mechanism of NLRP3 activation needs to be further investigated.

The regulatory effect of DON on the activity of NLRP3 inflammasomes is directly related to host immune status. Humans and animals are generally exposed to low doses of DON in the natural environment, and an exposure dose of less than 1 μg/kg BW does not cause harmful effects. When the host is in a state of pathogen infection, sensitivity to the DON dose increases. The production of NLRP3 inflammasomes, key regulators of inflammation, is also increased. Low-dose DON was able to increase pro-IL-1β, NLRP3, and caspase-1 in IPEC-J2 cells infected with ETEC K88 and enhance activation of NLRP3 inflammasomes [50]. Exposure to a low dose of DON (2 μM) enhanced NLRP3 inflammasome activation and induced pyroptosis-mediated toxicity. Surprisingly, a high dose of DON (16 μM) inhibited the chemotaxis and phagocytosis of macrophages [60]. Macrophages are a key link in inflammation leading to tissue damage. DON exposure aggravated PEDV-induced immunosuppression in a porcine alveolar macrophage model by inhibiting the TLR4/NF-κB/NLRP3 signaling pathway [59]. Sertoli cells are immune-privileged cells in the testes. Song et al. [61] reported that incubation with 10 μM DON significantly increased the expression of pyroptosis-associated genes for caspase-1 and GSDMD-N in Sertoli cells of *E. asinus*. Therefore, DON has a bidirectional effect on activation of NLRP3 inflammasomes. Whether high-level exposure to DON leads to immunosuppression via the NLRP3 inflammasome remains unclear.

### 3.2. AFB1 and the NLRP3 Inflammasome

Since the discovery of AFTs in the 1960s, knowledge of mycotoxins, which are toxic secondary metabolites secreted primarily by *Aspergillus* spp. (*A. flavus* and *A. parasiticus*), has increased dramatically. The percentage of animal feeds contaminated by AFTs was 47% in the Middle East and North Africa during the years from 2012 to 2020 [40] and 35–58% in animal feeds from Italy over 5 years (2018–2022) [41]. However, the detection rate of AFTs was 7–17% in 2023 in animal feeds from northern Spain [42]. AFTs are derivatives of coumarin and dihydrofuran with similar chemical structures. About 21 AFTs have been identified, with AFB1, AFB2, AFG1, and AFG2 being the most significant. The molecular formula of AFB1 ((3S,7R)-11-methoxy-6,8,19-trioxapentacyclo[10.7.0.02,9.03,7.013,17]nonadeca-1,4,9,11,13(17)-pentaene-16,18-dione) is C_17_H_12_O_6_. AFB1 is a very toxic AFT to humans and animals, and is listed as a group 1A carcinogen by the International Agency for Research on Cancer (IARC). AFB1 is frequently detected in a variety of nuts, particularly peanuts and walnuts, and has been shown to cause many ailments, including cancer, hepatitis, abnormal mutations, and other diseases. AFTs have good structural stability and can only be degraded at temperatures above 280 °C. AFB1 poisoning induces hepatotoxicity, enterotoxicity, nephrotoxicity, immunotoxicity, neurotoxicity, and reproductive toxicity in humans and animals, leading to substantial detrimental consequences. Most studies have confirmed the toxic effects of AFB1 in terms of apoptosis, autophagy, DNA damage, the inflammatory response, lipid peroxidation, lysosomal damage, mitochondrial dysfunction, necrosis, OS, and ROS production [62]. The signaling pathways associated with AFB1-mediated toxicity in mammalian cells are mainly the MAPK, NF-κB, NLRP3, Nrf2/ARE, p21, p53, PI3K/Akt/mTOR, TLR2/4, Wnt/β-catenin, and NLRP3/caspase-1 pathways [63].

The effect of AFB1 on NLRP3 inflammasome activity has been studied in vivo and in vitro. The liver is the primary detox organ and is also the main site of AFB hepatotoxicity. AFB1 induced NLRP3 inflammasome-mediated pyroptosis in hepatocytes through dephosphorylation of cyclooxygenase-2 (COX-2) [64]. AFB1 enhanced the expression of NLRP3 and pro-IL-1β by regulating NF-κB and downstream caspase-1 activity, p10, IL-1β, and GSDMD, in HepaRG cells and primary Kupffer cells (KCs). The enhanced proinflammatory signaling of KCs then activated the NLRP3 inflammasome in hepatocytes and upregulated the expression of COX-2 and proteins involved in assembly (NLRP3, ASC, and p10) and activation (IL-1β and GSDMD) of NLRP3 inflammasomes in primary hepatocytes [64]. Consistent with this report, Lv et al. [65] also studied the role of AFB1-induced severe pyroptosis-dependent hepatotoxicity through NLRP3/caspase-1/GSDMD signaling in AML12 cells.

In vivo, AFB1 induced pyroptosis by enhancing NLRP3 inflammasome assembly and activation (NLRP3, caspase-1 and GSDMD) and the release of IL-1β and IL-18 in the livers of BALB/c and C57BL/6 mice [66]. NLRP3-dependent pyroptosis in hepatocytes and liver injury in C57BL/6 mice were triggered by AFB1 via upregulation of COX-2 expression and an increase in mitochondrial damage [65,67]. Excessive ROS production activates the thioredoxin-interacting protein (TXNIP) upon dissociation from thioredoxin, which then binds to NLRP3 to promote inflammasome activation [68]. mtDNA synthesis induced by binding to TLRs is essential for activation of NLRP3 inflammasomes. TLR signaling triggers IRF1-dependent CMPK2 transcription and induces mtDNA synthesis through MyD88 and the TRIF adaptor, which activates the NLRP3 inflammasomes [69]. In addition, Ca^2+^-mediated mitochondrial damage can trigger the activation of NLRP3 inflammasomes [70]. PINK1/Parkin-mediated mitophagy is important in decreasing AFB1-induced liver injury in mice [62]. Therefore, the mechanism of how mitochondrial damage affects NLRP3 inflammasome activation via AFB1 needs further investigation. Moreover, AFB1-induced liver and splenic pyroptosis in mice can be mediated by disturbing the gut microbiota–immune axis [71,72].

In the duck liver, 60 µg/kg AFB1 induced acute liver damage by upregulating TXNIP, IL-18, NLRP3, and caspase-1 [73]. The Janus kinase 2 (JAK2)/signal transducer and activator of transcription 3 (Stat3) signaling pathway is closely associated with NLRP3 inflammasome activation. Furthermore, AFB1 significantly upregulated NLRP3, ASC, caspase-1, and GSDMD, thereby leading to pyroptosis and fibrosis mediated by JAK2/NLRP3 signaling [74]. In chicken liver, AFB1 significantly induced the expression of TNF-α, IL-6, IL-1β, iNOS, COX-2, NLRP3, caspase-1, -3, and -11, which resulted in immunotoxicity [75]. Mixed lineage kinase domain-like (MLKL) is a substrate of receptor-interacting protein kinase 3 (RIPK3) and induces the production of NLRP3/caspase-1 inflammasomes. The expression of TLR4/MyD88/NF-κB and necroptosis-related kinases RIPK1/RIPK3/MLKL in chicken liver was upregulated following a dose of 1 mg/kg AFB1 [76]. However, the role of the RIPK1/RIPK3/MLKL signaling pathway in NLRP3/caspase-1 inflammasome activation in AFB1-induced liver injury is still unclear. Additionally, AFB1 induces ileum damage in ducks by increasing TLR4, NF-κB, TNF-α, IL-6, TXNIP, NLRP3, and IL-18 [77]. Intestinal microbial dysbiosis caused by the upregulation of inflammatory factors after AFB1 exposure in mice may be related to the NLRP3 signaling pathway [78,79].

Furthermore, AFB1 induced GSDMD-mediated pyroptosis and inflammatory cytokine expression in primary microglia from C57BL/6 mice by activating the NLRP3 inflammasome [80]. In addition, 100 µg/kg AFB1 in C57BL/6J mice and 50 µM AFB1 in primary microglia inhibited the proliferation and neuronal differentiation of neural stem/precursor cells through upregulation of NLRP3, caspase-1, GSDMD-N, and IL-1β in the hippocampus [6]. In rats, AFB1 (0.15 and 0.3 mg/kg) caused increased OS and myocardial structural damage by activating the NLRP3 signaling pathway [81]. Therefore, AFB1 exposure causes various pathological changes in the host through the activation of NLRP3-mediated inflammation. Interestingly, chronic mild stress aggravated these changes in mice.

### 3.3. ZEA and the NLRP3 Inflammasome

ZEA is a harmful oestrogen-like *Fusarium* toxin primarily produced by *Fusarium* spp., including *F. graminearum, F. tricinctum*, *F. culmorum*, *F. equiseti*, *F. sernitectum*, and *F. solani*. ZEA is a well-studied mycotoxin (F-2 toxin). The percentage of animal feeds contaminated by ZEA was 33% in the Middle East and North Africa during the years from 2012 to 2020 [40] and 20–25% in animal feeds from Italy over 5 years (2018–2022) [41]. However, the detection rate of DON was 49–66% in 2023 in animal feeds from northern Spain [42] and 47.02% in 2024 in swine feed from South Korea [43]. The molecular formula of ZEA ((4S,12E)-16,18-dihydroxy-4-methyl-3-oxabicyclo[12.4.0]octadeca-1(14),12,15,17-tetraene-2,8-dione) is C_18_H_22_O_5_. Long-term exposure to ZEA causes severe toxic effects and high oestrogenic activity, including hepatotoxic, immunotoxic, and genotoxic effects. The mechanism of cell damage induced by ZEA seems to be more complex, mainly inducing apoptosis, arresting the cell cycle, damaging DNA, and leading to endoplasmic reticulum stress, inflammation, mitochondrial disruption, and OS. Swine are very susceptible to ZEA toxicity. It can cause severe reproductive disorders in female pigs, such as low fertility, abnormal fetal development, reduced litter size, and changes in reproductive hormone levels [82,83].

Studies have shown that the inflammatory response is the primary player in ZEA-induced intestinal toxicity, via interfering with the TLR4/NF-κB pathway [84,85]. In IPEC-J2 cells, ZEA exposure activated the ROS-mediated NLRP3 inflammasomes, which increased the secretion of caspase-1-dependent inflammatory factors and initiated the intestinal inflammatory cascade response [86,87]. Dextran sulfate sodium (DSS) is a chemical colitogen with anticoagulant properties that has been widely used to induce colitis. This model is especially helpful for delving into the role of innate immune mechanisms in intestinal inflammation because of its similarity to human ulcerative colitis. DSS increased the expression levels of NLRP3, ASC, caspase-1, pro-IL-1β, and pro-IL-18 in IPEC-J2 cells and in mouse colon tissues, consistent with the findings that ZEA induced caspase-1 activation via the NLRP3 inflammatory complex [84]. However, ZEA has a surprising degree of protection against inflammatory responses. The reason for this might be that the structure of ZEA is similar to that of oestradiol-17β [85]. After coadministration of DSS and ZEA, inflammatory cell infiltration and tissue damage were significantly restored. Oestradiol-17β is an ovarian oestrogen with anti-inflammatory properties, and its anti-inflammatory activity can be mediated through the oestrogen receptors ERα and ERβ. Intestinal ERβ expression has a protective effect on colitis-associated colorectal cancer. The expression of ERβ was extremely high in the colons of mice treated with DSS and ZEA [84]. Additionally, administration of up to 10 µM ZEA did not significantly upregulate ASC, IL-1β, or IL-18 expression in bovine primary theca cells; the expression levels of NLRP3 and IL-1β were weakly altered [57]. However, whether high-level exposure to ZEA activates the NLRP3 inflammasome in reproductive organs remains unclear [57]. Interestingly, ZEA suppressed the LPS-induced macrophage immune response by decreasing proinflammatory mediators and cytokines [88].

The autophagy-mediated regulation of the activity of NLRP3 inflammasomes is essential for immune homeostasis. In a rat insulinoma cell line, ZEA promoted activation of NLRP3 inflammasomes by modulating NF-κB/p65 signaling to induce NLRP3-dependent pyroptosis and inflammation. However, the increase in autophagy prevented ZEA-induced NLRP3 inflammasome activation and inflammatory responses [5].

### 3.4. T-2 Toxin and the NLRP3 Inflammasome

T-2 toxin is a prevalent A-subtype trichothecene mycotoxin secreted by *Fusarium* spp., including *F. acuminatum*, *F. equiseti*, *F. poae*, and *F. sporotichioides*. The percentage of animal feeds contaminated by T-2 toxin was 18% in the Middle East and North Africa during the years from 2012 to 2020 [40]. However, the percentages related to T-2 toxin contamination were 70–100% in animal feeds from Italy over 5 years (2018–2022) [41]. The molecular formula of T-2 toxin ([(1S,2R,4S,7R,9R,10R,11S,12R)-11-acetyloxy-2-(acetyloxymethyl)-10-hydroxy-1,5-dimethylspiro[8-oxatricyclo[7.2.1.02,7]dodec-5-ene-12,2′-oxirane]-4-yl] 3-methylbutanoate) is C_24_H_36_O_9_. Although T-2 toxin is not considered as a human carcinogen by the IARC, it causes hepatotoxicity, immunotoxicity, nephrotoxicity, neurotoxicity, and reproductive toxicity in multiple organs [89]. The harmful activity of T-2 toxin in the intestine includes OS, inflammatory responses, and apoptosis [90]. Male reproductive impairment is an important consequence of T-2 toxin toxicity through a variety of mechanisms, including causing testicular cell apoptosis, reducing sperm production and density, and increasing the deformity rate of sperm. Newborn or juvenile animals are more sensitive to T-2 toxin, and there is no specific prevention or treatment method for T-2 toxin poisoning.

Research has shown that IL-1β secretion induced by the fungal trichothecene mycotoxin roridin A is dependent on the NLRP3 inflammasome through P2X7R and Src tyrosine kinase signaling in human primary macrophages [91]. To date, there have only been two studies on the mechanisms by which T-2 toxin affects NLRP3 activation and pyroptosis. Yang et al. [92] reported that T-2 toxin induced ROS that activated NLRP3 inflammasomes and caused testicular inflammatory damage, thereby leading to disruption in TM4 cells and severe reproductive disorders in male mice [92]. Kidneys are the primary targets of T-2 toxin, which activated NLRP3 inflammasomes causing fibrosis via the mtROS/NLRP3/Wnt/β-catenin axis in a T-2 animal model [93].

### 3.5. OTA and the NLRP3 Inflammasome

OTA mycotoxins are widely found in several species of *Aspergillus* spp. including *A. ochraceus*, *A. carbonarius*, *A. niger*, *A. steynii*, *A. subramanianii*, and *A. westerdijkiae* and *Penicillium* spp. including *P. brevicompactum*, *P. chrysogenum*, *P. commune*, *P. cyclopium*, *P. nordicum*, *P. polonicum*, *P. verrucosum*, and *P. viridicatum*. OTA can contaminate a variety of foodstuffs, such as wheat, maize, and beans. The detection rate of OTA over a large range of values is 5–31% in animal feeds [42,43]. The molecular formula of OTA ((2S)-2-[[(3R)-5-chloro-8-hydroxy-3-methyl-1-oxo-3,4-dihydroisochromene-7-carbonyl]amino]-3-phenylpropanoic acid) is C_20_H_18_ClNO_6_. Due to its stable chemical properties and strong tolerance of OTA, it easily survives in feed and retains its strong toxicity. OTA has a half-life of more than a month in humans and animals. It has now been shown that OTA conveys strong nephrotoxic, hepatotoxic, immunotoxic, carcinogenic, and teratogenic effects. OTA is listed as a group 2B carcinogen by the IARC. As kidneys are the target organ of OTA, renal injury, renal fibrosis, interstitial nephritis, and other nephropathies are common in the kidneys of rodents, pigs, and humans [94]. OTA induces nephrotoxicity both in vivo and in vitro by promoting apoptosis, inducing OS, regulating autophagy, and inhibiting mitosis [95]. 

However, few investigations have concentrated on the interaction between OTA and NLRP3 inflammasome-mediated pyroptosis. Madin–Darby canine kidney (MDCK) cells incubated with OTA exhibited significant activation of NLRP3 inflammasomes, caspa-se-1-dependent pyroptosis, and upregulation of IL-1β, IL-6, IL-18, TNF-α, and GSDMD [4]. Furthermore, OTA caused pyroptosis in PK-15 cells by increasing levels of NLRP3, GSDMD, caspase-1 p20, ASC, pro-caspase-1, and IL-1β, leading to nephrotoxicity [96]. Li et al. [4] first revealed that NLRP3 inflammasome-mediated pyroptosis was associated with OTA-induced nephrotoxicity. The expression of pyroptosis-related proteins such as NLRP3, caspase-1, GSDMD, and IL-1β was elevated in the kidneys of C57BL/6 mice intraperitoneally injected with OTA. OTA exposure caused intestinal aging damage in both murine intestinal tissues and the rat small intestine crypt epithelial cell line IEC6 through the NLRP3 signaling pathway. The mechanism is related to Ca^2+^ overload, mitochondrial OS, and inflammation [97].

In addition, studies have reported that mitochondrial damage is one of the mechanisms of OTA-induced cytotoxicity in various cell lines [98,99]. It has been shown that OTA can lead to disruption of mitochondria and cell death via apoptosis and autophagy in human gastric epithelial cells [100]. Additionally, OTA upregulated the expression of TFAM (an mtDNA transcription factor) in ducklings and enhanced mitochondrial OS in the jejunal mucosa [101]. Mitochondrial damage is an important pathway activating NLRP3 inflammasomes. Whether OTA induces the activation mechanism of NLRP3 inflammasomes through mitochondrial damage remains to be further studied.

### 3.6. FB1 and the NLRP3 Inflammasome

To date, 28 analogues for fumonisins have been found, including 11 kinds of FBs, in which FB1 is the main component [98]. The percentage of animal feeds contaminated by fumonisins was 47% in the Middle East and North Africa during the years 2012–2020 [40] and 20–40% in animal feeds from Italy over the 5 years 2018–2022 [41]. However, the detection rate of fumonisins was 68.31% in swine feed from South Korea [43]. FB1 is a water-soluble metabolite produced by *Fusarium* spp., such as *F. moniliforme*, *F. verticillioides*, *F. oxysporum*, *F. nygamai*, *F. fujikuroi*, and *F. proliferatum*. FB1 occurs naturally in many agricultural crops. The molecular formula of T-2 toxin (2,2′-{(19-amino-11,16,18-trihydroxy-5,9-dimethylicosane-6,7-diyl)bis[oxy(2-oxoethane-2,1-diyl)]}dibutanedioic acid)) is C_34_H_59_NO_15_. FB1 is considered as a group 2B carcinogen by the IARC, with strong systemic toxicity, including hepatotoxicity, neurotoxicity, nephrotoxicity, reproductive toxicity, and immunotoxicity [99]. Autophagy, apoptosis, endoplasmic reticulum stress, and OS each play dual roles in FB1-induced toxicity.

Ma et al. found that FB1 caused pyroptosis in IPEC-J2 cells by upregulating preprotein translocation factor (Sec62) to activate the PKR-like ER kinase protein pathway [102]. FB1 also promoted inflammatory cytokine secretion and upregulated the expression of NLRP3 and caspase-1, causing damage in IPEC-J2 cells and in intestinal epithelial cells in mice [103]. However, FB1-induced NLRP3-dependent pyroptosis is shielded by autophagy. FB1-induced pyroptosis increases the level of autophagy to a certain extent, which is related to the inhibition of mechanistic target of rapamycin (mTOR) phosphorylation. The mTOR inhibitor, rapamycin, inhibited the expression of NLRP3 and downregulated FB1-induced pyroptosis in IPEC-J2 cells, indicating the capacity to alleviate intestinal inflammatory injury. This indicates that increasing the level of autophagy may be used as a method to against tissue damage induced by FB1 exposure in practice.

In addition to the abovementioned mycotoxins, patulin (PAT) can activate NLRP3 inflammasomes [104]. PAT (4-hydroxy-4H-furan [3, 2c] pyran-2 [6H]-ketone) is a common mycotoxin in fruits and vegetables, and its molecular formula is C_7_H_6_O_4_. Although PAT is considered a noncarcinogen, it is genotoxic, cytotoxic, and neurotoxic, mainly causing liver and kidney damage. PAT also induced pyroptosis and inflammatory damage through activation of NLRP3 inflammasomes in mouse livers and HepG2 cells [104]. Mycophenolic acid (MPA) is an in vivo metabolic product of mycophenolate mofetil. Although MPA alone does not affect the levels of NF-κB p-p65 or pro-IL-18 in THP-1 cells, it synergizes with LPS to greatly promote the secretion of IL-18 through activation of NLRP3 inflammasomes [105]. However, it is unknown whether other mycotoxins, such as *Alternaria* toxins, citrinin, enniatins, ergot alkaloids, nivalenol, and HT-2 toxin, are capable of triggering NLRP3 inflammasome activation.

## 4. Targeting the NLRP3 Inflammasome for Mycotoxin Exposure

Much research on pharmacological interference with the activation of NLRP3 in-flammasomes has been recently published [33]. Theoretically, common inflammasome components or signaling molecules activating NLRP3 inflammasomes could be therapeutic targets for inhibiting NLRP3-dependent inflammation. To our delight, the development of NLRP3 inhibitors has been vigorously promoted, and a range of NLRP3 inhibitors have been discovered and have shown promising therapeutic potential in clinical trials with a wide array of autoinflammatory and chronic inflammatory diseases [33]. Mycotoxin-induced nephrotoxicity, hepatotoxicity, and immunotoxicity are closely related to the NLRP3 inflammasome, making it a likely target for pharmacotherapy. Inhibiting the activation of NLRP3 inflammasomes can effectively reduce the toxic effects of mycotoxins. Therefore, targeting NLRP3-mediated inflammation represents a novel therapeutic option for against mycotoxin-induced toxicity [33,106,107].

MCC950 (originally described as CRID3 and CP-456,773), a diaryl sulfonylurea-containing compound, was first identified by Gabel et al. [34]. MCC950 is the best-studied small-molecule inhibitor of NLRP3 and does not affect the NLRP1, NLRC4, or AIM2 inflammasome- or TLR-mediated priming signals. MCC950 directly interacts with the Walker B motif in the NACHT domain of NLRP3, thereby preventing ATP hydrolysis, NLRP3 activation, and inflammasome generation [108]. MCC950 has since been widely studied for its therapeutic effects on autoimmune, cardiovascular, metabolic, and other diseases and has entered phase-II clinical trials [109]. MCC950 has also been shown to attenuate DON-induced pyroptosis and inflammatory responses by preventing activation of NLRP3 inflammasomes. In TM4 cells, inflammatory damage and upregulation of NLRP3, ASC, caspase-1, IL-1β, and IL-18 caused by 4 nM T-2 toxin were markedly lower in the presence of 20 nM MCC950 [92]. However, MCC950 (20 nM) did not lower ROS levels in TM4 cells [92]. The fibrosis caused by T-2 toxin and the structural and functional damage through activation of NLRP3 inflammasomes and the Wnt/β-catenin pathway were mitigated by MCC950 in an HK-2 cell model (10 μM) and a C57BL/6N mouse model (10 mg/kg) [93]. MCC950 (10 μM) blocked FB1-induced pyroptosis by abolishing the expression of GSDMD and the release of proinflammatory factors, IL-1β and IL-18, induced by the combination of FB1 and DON in IPEC-J2 cells [58,103]. The above phenomena have also been observed in DON-treated IPEC-J2 cells and HaCaT cells [48,54] and in patulin-treated HepG2 cells. Therefore, MCC950 can serve as a potential treatment for alleviating toxicity-induced mycotoxins [104].

BAY 11-7082 is a phenyl vinyl sulfone compound, and its chemical structure is (E)-3-[4-methylphenylsulfonyl]-2-propenenitrile). BAY 11-7082 specifically inhibits NF-κB, blocks the activation and translocation of NF-κB from the cytoplasm to the nucleus, and shows many pharmacological actions, such as anticancer, neuroprotective, and anti-inflammatory effects. Recent research has shown that BAY11-7082 can inhibit cell pyroptosis by covalently modifying the Cys site of GSDMD, thereby inhibiting the membrane perforation process. In IPEC-J2 cells, Ge et al. [49] reported that BAY 11-7082 (2.5 μM) inhibited NLRP3 and procaspase-1 expression and reversed the inflammation induced by the binding of DON to LPS. BAY 11-7082 (10 μM) blocks 0.05 μg/mL AFB1-promoted swine influenza virus replication and inflammatory responses in 3D4/21 cells by inhibiting NF-κB signaling [110]. BAY 11-7082 (10 μM) also inhibited the inflammatory responses induced by a combination of AFB1 (0.16 mg/mL) and OTA (0.4 mg/mL) in 3D4/21 cells [111]. The NF-κB signaling pathway is a priming step in NLRP3 inflammatory responses. In 0.3 μg/mL AFB1-treated AML12 cells, BAY 11-7082 (10 μM) attenuated the upregulation of inflammatory cytokines and NF-κB/p65 activation [112].

Curcumin (1,7-bis-(4-hydroxy-3-methoxyphenyl)-1,6-heptadiene-3,5-dione) is the active compound in *Curcuma longa* (turmeric) and has multiple biological activities, including antibacterial, anticancer, anti-inflammatory, anti-lipidaemic, antioxidant, antiviral, cardioprotective, immunoregulatory, and neuroprotective effects. Curcumin specifically inhibits NLRP3 inflammasomes by blocking potassium efflux and interfering with mitochondrial transport and ASC polymerisation but not on the AIM2 or NLRC4 inflammasomes. Several researchers have demonstrated that curcumin can significantly reduce AFB1-mediated liver and intestinal damage and renal dysfunction, both in vitro and in vivo. Curcumin was able to reverse AFB1-induced intestinal injury in ducks by restoring the intestinal epithelial barrier, reducing inflammation and pyroptosis, and balancing intestinal microbiota via modulating NLRP3 inflammasomes and the TLR4/NF-κB signaling pathway [79]. Curcumin (500 mg/kg) alleviated AFB1-induced liver pyroptosis and acute liver damage by regulating the JAK2/NLRP3 signaling pathway [74] and NLRP3/caspase-1 signaling pathways in ducks [73]. Curcumin (500 mg/kg) also protected the ileum against acute damage in ducks (*Anas platyrhynchos*) by activating Nrf2-ARE signaling and inhibiting NLRP3/caspase-1 and NF-κB signaling [77]. Curcumin (100 and 200 mg/kg) inhibited 0.75 mg/kg AFB1-induced pyroptosis in the livers of BALB/c mice by inhibiting NLRP3 inflammasome assembly [113]. Proteomic profiling revealed that curcumin (100 mg/kg) effectively prevented 0.75 mg/kg AFB1-induced pyroptosis in mice with liver injury [66].

Luteolins (3,4,5,7-tetrahydroxy flavones) are flavonoids mainly found in fruits, vegetables, and herbs. Luteolin has multiple useful effects, including antitumour, anti-inflammatory, antiviral, antioxidant, and immune regulatory effects. Luteolin inhibits NLRP3 inflammasome activation by preventing the interaction of NLRP3 with ASC. To date, the protective effect of luteolin on inflammatory injury induced by mycotoxin exposure following NLRP3 inflammasome activation has not been directly reported. However, luteolin mitigated DON-induced toxicity in broilers by reducing oxidative damage [114] and alleviated AFB1-induced apoptosis and OS in mouse [115] and rat liver [116]. Luteolin (50 and 100 μg/mL) also reduced OTA-induced damage on Vero cells and rat lymphocytes [117]. Luteolin reduced OTA-induced OS by reducing ROS levels, reversing the reduction in mitochondrial membrane potential, activating antioxidant enzymes, and regulating the Nrf2 and HIF-1α pathways in NRK-52E rat kidney cells (100 μM) and IPEC-J2 cells (8.7 μM) [118,119]. Luteolin (8, 16, and 32 μg/mL) also reduced FB1-induced damage to the intestine from inflammation, through inhibiting NF-κB and ERK signaling and reducing IL-6 and IL-1β expression in IPEC-J2 cells [120]. It has been reported that luteolin could serve as a target for limiting mycotoxin-induced neuroinflammation and improving neuropsychiatric diseases [121]. Thus, luteolin could alleviate mycotoxin-induced toxicity through a reducing OS and ameliorating the reduction in the mitochondrial membrane potential.

Nonsteroidal anti-inflammatory drugs (NSAIDs) can reduce inflammation, acute fever, and pain. Piroxicam is an NSAID that suppresses NLRP3 activation via reversible inhibition of volume-regulated anion (chloride) channels and the NF-κB pathway. To date, the mitigating effect of NSAIDs on inflammatory injury induced by mycotoxin exposure following NLRP3 inflammasome activation has not been described. Interestingly, piroxicam (5 mg/kg/48 h for 14 d) proved useful in preventing the chronic toxic effects of OTA (289 μg/mL), particularly nephrotoxicity, in rats [122]. However, indomethacin, phenylbutazone, and aspirin were not effective in preventing mortality in mice administered with T-2 toxin [123]. Therefore, whether NSAIDs can alleviate mycotoxin toxicity via inhibiting the NLRP3 inflammasome requires further research.

Tanshinone I (Tan I) and Tan IIA are the active ingredients in *Salvia miltiorrhiza* plants. Tanshinone exerts anticancer, antioxidant, and anti-inflammatory effects via different mechanisms. Tan I prevents the formation and activation of NLRP3 inflammasomes by blocking the NLRP3-ASC interaction. To date, there has been no report that Tan I can reduce the inflammatory injury from mycotoxin exposure following NLRP3 inflammasome activation. However, Tan IIA (45 µg/mL) did show a protective effect against DON-mediated toxicity in IPEC-J2 cells by inhibiting the expression of NLRP3, caspase-1, GSDMD, IL-1β, and IL-18 [47] and restoring mitochondrial function via quality control [124].

Licochalcone, a major flavonoid compound in liquorice, contains bioactive components denoted as A-G. These bioactive components exhibit various biological activities and anti-inflammatory effects, including antiparasitic, antibacterial, antioxidant, anti-inflammatory, antitumorigenic, and anticancer effects. Lico A (0.925, 1.85, and 3.7 μM) significantly mitigated damage from inflammatory mediators, OS, and pyroptosis, which inhibited TLR4/NF-κB/MAPK and NLRP3/caspase-1/GSDMD signaling pathway activation in AML12 cells. In vivo, Lico A (5 mg/kg) ameliorated AFB1-induced hepatotoxic effects in C57BL/6 mice via the inhibition of inflammation and GSDMD-mediated pyroptosis [65].

Autophagy negatively regulates NLRP3 inflammasome activation, inhibits the inflammatory response, and reduces inflammatory damage [125]. The mechanism by which autophagy inhibits NLRP3 inflammasomes may be related to a reduction in the expression of ASC protein, NLRP3 phosphorylation, and the clearance of mtROS. Hemin and its derivative, cobalt protoporphyrin, inhibited assembly of NLRP3 inflammasomes by enhancing autophagy, resulting in increased degradation of ASC [126]. Phosphorylation of NLRP3 promoted entry of the NLRP3 protein into autophagosomes and inhibiting the activation of NLRP3 inflammasomes [127]. Rapamycin, an inhibitor of mTOR, reduced NLRP3 inflammasome activation by attenuating the mTOR/NF-κB signaling pathway in macrophages [128]. Notably, rapamycin (50 nM) decreased damage to the intestine from FB1-induced inflammation and inhibited the upregulation of pyroptosis-related genes in IPEC-J2 cells [103]. Injection (i.p.) of 5 mg/kg rapamycin protected animals from FB1-created enteritis by inhibiting the expression of pyroptosis-related genes. PINK1/Parkin-induced mitophagy alleviated mitochondrial damage from exposure to T-2 toxin by decreasing the activation of NLRP3 inflammasomes [129]. Another study showed that PINK1/Parkin activation alleviated AFB1-induced liver injury, and that Parkin expression deficiency aggravated NLRP3 inflammasome activation and mitochondrial damage in AFB1-exposed mice [67]. However, 3-methyladenine (1 mM), an autophagy inhibitor, was shown to attenuate the LDH release, activation of NLRP3 inflammasomes, pyroptosis, and inflammatory responses induced by patulin [104]. Therefore, the role of autophagy during pyroptosis needs to be verified through further experiments [125].

*Bacillus licheniformis* (2 μL/g) treatment effectively attenuated gut-testis axis damage caused by AFB1 exposure in mice through the NLRP3 signaling pathway [78]. Blocking GSDMD activity in AFB1-treated primary microglia from C57BL/6 mice with dimethyl fumarate (100 μM) reduced the secretion of IL-1β and IL-18 [80]. Melatonin (5 mg/kg) reduced NLRP3 inflammasome activation by reducing OS, thereby protecting against damage from 0.3 mg/kg AFB1-induced myocardial toxicity in rats [81]. Chlorogenic acid is a phenolic acid synthesized from caffeic acid and quinic acid that has anti-inflammatory, antioxidant, and damage repair properties. It was reported that chlorogenic acid (25 and 50 μg/mL) attenuated DON-induced dermal damage in HaCaT cells by inhibiting the expression of pyroptosis-related proteins, including NLRP3, cleaved caspase-1, GSDMD-N, and cleaved IL-1β, and IL-18 [54]. Microalgal astaxanthin (96 μM) mitigated the detrimental effects of DON-induced pyroptosis by modulating the mtROS/NF-κB-dependent activation of NLRP3 inflammasomes in lymphocytes from carp spleen [53]. The OTA-induced activation of NLRP3 inflammasomes is associated with the generation of ROS. Its nephrotoxicity can be alleviated by taurine; 1, 2, and 5 mM inhibited ROS production and changed the activity of antioxidant enzymes [96]. Selenomethionine (8 μM) significantly mitigated the cytotoxicity from the incubation of MDCK cells with OTA (2.0 μg/mL) by inhibiting the activation of NLRP3 inflammasomes and NLRP3/caspase-1-dependent pyroptosis [130]. 

The above studies reveal that inhibiting the activation of NLRP3 inflammasomes can significantly reduce the toxic effects of mycotoxins (Table 1). In the future, we will focus on identifying NLRP3 inhibitors that alleviate the injurious effects of fungal toxins.

## 5. Conclusions and Future Perspectives

Mycotoxins are harmful compounds synthesized by specific filamentous fungi that pose a danger to human and animal health. Contamination with mycotoxins causes wast-age of food and animal feed and harms the world’s commerce in farm crops. The control of mycotoxins is based on two strategies: prevention of toxin production and detoxification. Biodegradation of mycotoxins as a detoxification method has become a popular subject for research. The use of specific microbes to reduce mycotoxin contamination of crops is highly effective and broadly applicable. In recent years, our group has screened and identified bacteria with the capability of digesting mycotoxins including ZEA-degrading *Bacillus velezensis* [131], AFB1-degrading *Bacillus subtilis* [132], and DON-degrading *Bacillus cereus* [133]. Probiotics are safe and effective alternatives to fungicides for controlling mycotoxin contamination, but the toxicological mechanisms of mycotoxins are complex and unclear and need to be further studied to clarify their pathogenic activity.

The innate inflammatory immune response protects the body from pathogenic attack, but chronic inflammation can lead to diseases such as cancers. In recent years, the toxicological mechanisms of mycotoxins, including DON, AFB1, ZEA, T-2 toxin, OTA, and FB1, have been associated with the systemic NLRP3 inflammasome, which contributes to immunotoxicity, hepatotoxicity, intestinal toxicity, and neurotoxicity. Here we have summarized the recent research on the dysregulated expression of the NLRP3 inflammasome in response to mycotoxin exposure and the roles of NLRP3 inflammasome inhibitors in alleviating tissue damage and the inflammatory response induced by mycotoxins.

(1)As mentioned above, mycotoxins trigger NLRP3 inflammasome activation via the canonical activation pathway. Although evidence suggests that exogenous stimuli such as DON, AFB1, ZEA, OTA, T-2 toxin, FB1, or patulin can trigger NLRP3 inflammasome activation, whether these mycotoxins act directly on NLRP3 remains unclear. Previous studies have investigated alterations in the expression of factors involved in generation of NLRP3 inflammasomes in response to mycotoxin exposure, but the exact mechanisms and signaling pathways by which mycotoxins induce NLRP3 inflammasome activation have remained elusive.(2)Most of the research on NLRP3 inflammasome activation has focused on DON, AFB1, and ZEA, whereas studies of OTA, T-2 toxin, FB1, mycophenolic acid, and patulin are scarce; alternaria, HT-2 toxins, citrinin, enniatins, ergot alkaloids, and nivalenol have not been studied at all. Based on the existing research, it is currently not possible to ascribe a set of common factors to the activation of NLRP3 inflammasomes triggered by all 500 mycotoxins. Therefore, a wide range of mycotoxins as a subject of future study is necessary to improve understanding of the general rules for mycotoxin-induced NLRP3 inflammasome activation.(3)In recent years, more than 20 pharmacological inhibitors of NLRP3 inflammasomes have been reported, some of which exhibit promising therapeutic potential for treating NLRP3-related diseases in the clinic. Undeniably, all studies have clearly indicated that MCC950, BAY 11-7082, curcumin, and luteolin ameliorate tissue damage and inflammatory responses in models of mycotoxin exposure. Theoretically, any molecule or signal involved in the activation of NLRP3 inflammasomes, such as active caspase-1, GSDMD cleavage, proteins involved in inflammasome assembly, and inflammatory cytokines, can potentially inhibit the NLRP3 inflammasome. These are the two major mechanisms of small molecule inhibitors: they directly interact with the NLRP3 protein and bind to the ATP-binding motifs of the NACHT domain of NLRP3, subsequently inhibiting ATPase activity. In addition to the inhibitors mentioned above, other kinds of NLRP3 inflammasome inhibitors should be designed to investigate their effects on mycotoxin exposure. It will be worth examining whether the occurrence of chemical reactions between these inhibitors and mycotoxins reduces their toxic effects. For example, the carbon-carbon double bonds in BAY 11-7082 can react with the amino group in FB1 via Michael addition, and the sulfonyl group in MCC950 can react with the carboxyl group in FB1 via amide bond-forming reactions.

In summary, the activation of NLRP3 inflammasomes by mycotoxins via oxidative stress and the NF-κB pathway exacerbates the pathological conditions of the host. Our data improve the understanding of mycotoxin toxicity by demonstrating that the activation of NLRP3 inflammasomes has a critical role in the toxic effects of mycotoxins. Therefore, targeting the NLRP3 inflammasome could be an effective strategy for alleviating my-cotoxin-induced toxicity. However, there is still a long way to go before exploiting NLRP3 inflammasome inhibitors can be confidently applied to the treatment of disease. Future research should focus on the development of safe, specific, efficient, stable, nontoxic, inexpensive, and simple preparation processes for NLRP3 inflammasome inhibitors. We believe that the continuing advancement in biomolecular medicine and related technologies will result in a substantial number of new NLRP3 inflammasome inhibitors that will steadily be applied clinically to mitigate mycotoxin-induced toxicity and treat NLRP3-related inflammatory diseases.

## Figures and Tables

**Figure 1 vetsci-11-00291-f001:**
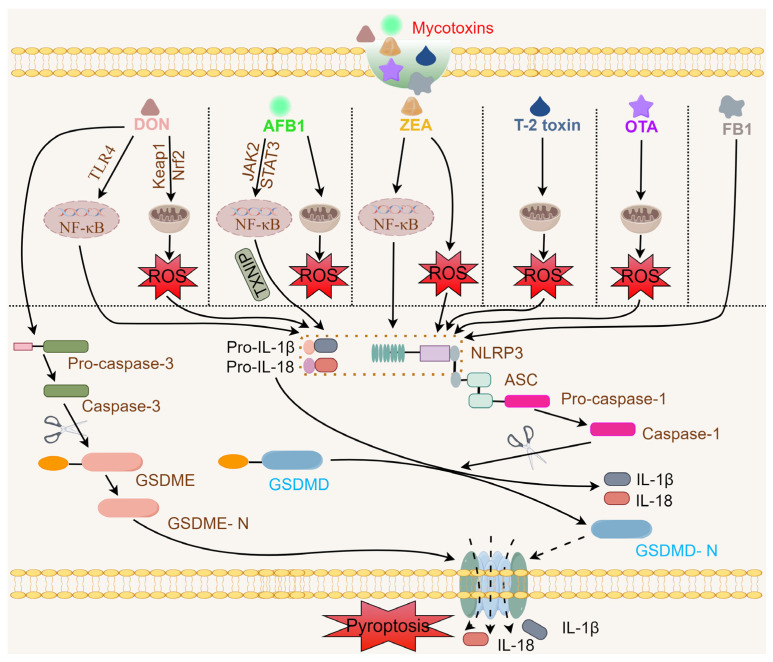
Depiction of activation of an NLRP3 inflammasome by mycotoxin (DON, AFB1, ZEA, OTA, and FB1) exposure (created in Figdraw). In the case of DON exposure, the first signal comes from TLR4 and NF-κB and increases the intracellular transcripts of pro-IL-1β, pro-IL-18, and NLRP3. The NLRP3 inflammasome is activated by NLRP3 oligomerization with ASC and pro-caspase-1. Caspase-1 cleavage of GSDMD forms active N-termini, causeing cell membrane perforation. Active caspase-1 then converts pro-IL-1β and pro-IL-18 to their biologically active structures, causing the release of IL-1β and IL-18 into the extracellular matrix, amplifying inflammation, and inducing pyroptosis. DON, deoxynivalenol; AFB1, aflatoxin B1; ZEA, zearalenone; OTA, ochratoxin A; FB1, fumonisim B1; TLR4, toll-like receptor 4; Keap1, kelch-like ECH-associated protein 1; Nrf2, nuclear factor erythroid-derived 2-like 2; JAK2, Janus kinase 2; STAT3, signal transducer and activator of transcription 3; TXNIP, thioredoxin interacting protein; ROS, reactive oxygen species; NF-κB, nuclear factor kappa-light-chain-enhancer of activated B cells; NLRP3, nucleotide-binding, oligomerization domain (NOD)-like receptor (NLR) family pyrin domain containing 3; ASC, apoptosis-associated speck-like protein containing a C-terminal caspase recruitment domain; GSDMD, gasdermin D; GSDMD-N, gasdermin D N-terminal; GSDME, gasdermin E; GSDME-N, gasdermin E N-terminal; IL, interleukin.

**Table 1 vetsci-11-00291-t001:** Literature summary of the effects of NLRP3 inflammasome inhibitors on mycotoxin exposure.

Inhibitors	Mycotoxins	Inhibitory Mechanism	Events	Ref.
MCC950	OTA, T-2 toxin, FB1, and DON	Binds to Walker B motif of NATCH domain to inhibit ATPase activity and close active conformation.	↓: NLRP3, pro-caspase-1, caspase-1, GSDMD, ASC, IL-18, pro-IL-1β, IL-1β, TNF-α, IL-6, NLRP3 inflammasome, and pyroptosis.	[4,48,54,58,92,93,103]
BAY 11-7082	DON, AFB1, OTA	Binds with NATCH and leucine-rich repeat domain and inhibits ATPase.	↓: NLRP3, pro-IL-1β, caspase-1, NF-κB signaling pathway, inflammatory responses, and cytokines.	[49,110,111,112]
Curcumin	AFB1	Prevents K^+^ efflux, inhibits microtubule-driven recruitment of ASC on mitochondria to NLRP3 on the endoplasmic reticulum.	↓: NLRP3 inflammasome, NLRP3/caspase-1 signaling pathways, TLR4/NF-κB signaling pathway, JAK2/NLRP3 signaling pathway, OS, inflammatory response, ITPR2, caspase-12/caspase-3 pathway, intestinal injury, fibrosis, and pyroptosis; ↑: Nrf2-ARE signaling pathway.	[66,74,77,79,113]
Luteolin	DON, AFB1, FB1, and OTA	Inhibits NLRP3 inflammasome activation by disrupting the interaction between NLRP3 and ASC.	↓: apoptosis, OS, NF-κB, ERK signaling pathways, IL-6, IL-1β, extracellular H_2_O_2_, intracellular ROS, DNA damage, toxicity, and inflammatory injury; ↑: antioxidant enzymes, Nrf2 and HIF-1α pathways.	[114,115,116,117,118,119,120]
Piroxicam	OTA	Suppresses NLRP3 activation via reversible inhibition of volume-regulated anion (chloride) channels and the NF-κB pathway.	↓: nephrotoxicity.	[122]
Tanshinone IIA	DON	Inhibits mitochondrial ROS release.	↓: NLRP3, caspase-1, GSDMD, IL-1β, IL-18, and cell injury; improved mitochondrial function via mitochondrial quality control.	[47,124]
Licochalcone A	AFB1	Binds to NEK7 and disrupts NEK7-NLRP3 interaction.	↓: TLR4-NF-κB/MAPK, NLRP3/caspase-1/GSDMD signaling pathway, oxidative insults, inflammation, pyroptosis, and hepatotoxicity.	[65]
Rapamycin	FB1	Downregulates NF-κB signaling pathway.	↓: intestinal inflammatory injury and pyroptosis-related genes.	[103]
MitoQ	T-2 toxin	Reduces mitochondrial DNA damage.	↓: mtROS, NLRP3-inflammasome, W/β signaling, structural and functional damage, and fibrosis	[93]
*Bacillus licheniformis*	AFB1	Inhibits the expression of the NLRP3 inflammasome.	↓: gut-testis axis damage and NLRP3-mediated NLRs signaling.	[78]
Dimethyl fumarate	AFB1	Downregulates NF-κB signaling pathway.	↓: inflammatory cytokines, pyroptosis, and neurotoxicity.	[80]
Melatonin	AFB1	Inhibits the expression of the NLRP3 inflammasome.	↓: NLRP3, ASC, caspase-1 p20, IL-1βp17, and myocardial toxicity.	[81]
Chlorogenic acid	DON	Inhibits the expression of the NLRP3 inflammasome.	↓: OS, inflammation, apoptosis, and MAPK/NF-κB/NLRP3 pathway; ↑: Nrf2/HO-1 pathway.	[54]
Microalgal astaxanthin	DON	Modulates the mtROS/NF-κB-dependent NLRP3 inflammasome.	↓: mtROS-NF-κB-dependent NLRP3 inflammasome and pyroptosis.	[53]
Taurine	OTA	Inhibits the production of ROS.	↓: pyroptosis and ROS; ↑: antioxidant enzymes.	[96]
Selenomethionine	OTA	Inhibits the expression of the NLRP3 inflammasome.	↓: NLRP3 inflammasome, NLRP3-caspase-1-dependent pyroptosis, and cytotoxicity.	[130]

OTA, ochratoxin A; FB1, fumonisim B1; DON, deoxynivalenol; NLRP3, nucleotide-binding, oligomerization domain (NOD)-like receptor (NLR) family pyrin domain containing 3; GSDMD, gasdermin D; ASC, apoptosis-associated speck-like protein containing a C-terminal caspase recruitment domain; IL, interleukin; TNF-α, tumor necrosis factor α; AFB1, aflatoxin B1; NF-κB, nuclear factor kappa-light-chain-enhancer of activated B cells; TLR4, Toll-like receptor 4; JAK2, Janus kinase 2; OS, oxidative stress; ITPR2, inositol 1,4,5-trisphosphate receptor, type 2; Nrf2, nuclear factor erythroid 2-related factor 2; ARE, antioxidant response element;ROS, reactive oxygen species; HIF-1α, hypoxia inducible factor-1α; MAPK, mitogen-activated protein kinase; NEK7, NIMA related kinase 7; MitoQ, mitoquinone; mtROS, mitochondrial ROS; HO-1, heme oxygenase-1; ↑ represents upregulated levels; ↓ represents downregulated levels.

## Data Availability

Not applicable.

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
