# Peer review of "The Novel Role of the NLRP3 Inflammasome in Mycotoxin-Induced Toxicological Mechanisms"

_vetsci, 2024, doi:10.3390/vetsci11070291_

Round 1
Reviewer 1 Report
Comments and Suggestions for Authors
This review article approaches the molecular mechanism and pathways of several mycotoxins like (AFB1, ZEA, DON, FB1, OTA and TH-2). These mechanisms can explain some toxicological effects, like immunosuppression and chronic inflammation, related to mycotoxicoses. The article is a good contribution to the field of mycotoxicology.
Please find in the attached file some minor revisions, which must be approached to consider this manuscript for publication in Veterinary Science.

Author Response
This review article approaches the molecular mechanism and pathways of several mycotoxins like (AFB1, ZEA, DON, FB1, OTA and TH-2). These mechanisms can explain some toxicological effects, like immunosuppression and chronic inflammation, related to mycotoxicoses. The article is a good contribution to the field of mycotoxicology.
Please find in the attached file some minor revisions, which must be approached to consider this manuscript for publication in Veterinary Science.
Response: Thank you very much for your evaluation of the paper. We are very grateful for your positive and constructive comments and suggestions. Those suggestions and comments are all valuable and very helpful for revising and improving our paper. We tried our best to improve the quality of the manuscript. Some mistakes and errors, including but not limited to your suggestions, were modified in the revised manuscript.
- L95, delete ‘and have good tolerance’.
Response: Thank you very much for your careful checks and suggestions. We have deleted the words ‘and have good tolerance’ in the revised manuscript. Please refer to the revised manuscript in the “2. Mycotoxicosis” section (The first paragraph).
- L96, delete ‘and strong adaptability to the environment’.
Response: Thank you very much for your careful checks and suggestions. We have deleted the words ‘and strong adaptability to the environment’ in the revised manuscript. Please refer to the revised manuscript in the “2. Mycotoxicosis” section (The first paragraph).
- L98, it is not true at all, please read the cited article and correct the sentence: "However, in those events where the concentration exceedance of the statutory limits (e.g. 20% or 25% of the samples) would lead to regular high dietary exposures to mycotoxins, and to their mixtures in particular, adverse health consequences can be expected. It is also of immense importance that the detectable levels (based on our estimation up to 60–80% of the samples) are not overlooked as through diets humans and animals are exposed to mixtures of different mycotoxins."
Response: Thank you very much for your careful checks and suggestions. We are very sorry for our incorrect writing. According to your suggestion, the sentence “The Food and Agriculture Organization of the United Nations has noted that approxi-98 mately one-quarter of crops worldwide are threatened by mycotoxin contamination [12].” have been corrected to “As noted by Eskola et al., up to 60-80% of the food crop samples worldwide are threatened by mycotoxin contamination [15].” in the revised manuscript. Please refer to the revised manuscript in the“2. Mycotoxicosis” section (The first paragraph).
- L99, delete ‘Some mycotoxins have acute toxicity.’.
Response: Thank you very much for your careful checks and suggestions. We have deleted the sentence ‘Some mycotoxins have acute toxicity.’ in the revised manuscript. Please refer to the revised manuscript in the “2. Mycotoxicosis” section (The first paragraph).
- L191, Nuclear factor kappa-light-chain-enhancer of activated B cells.
Response: Thank you very much for your careful checks and suggestions. We are very sorry for our incorrect writing. According to your suggestion, the words “nuclear factor 191 kappa B” have been corrected to “Nuclear factor kappa-light-chain-enhancer of activated B cells.” in the revised manuscript. Please refer to the revised manuscript in the “Figure 1”.
- L655, this sign (ihnibition, depletion) must be before the effects not after. Correct this in all the table.
Response: Thank you very much for your careful checks and suggestions. ↑ represents upregulated levels; ↓ represents downregulated levels. According to your suggestion, we have placed these signs before the effects in the revised manuscript. Please refer to the revised manuscript in the “Table 1”.
- L660, Nuclear factor kappa-light-chain-enhancer of activated B cells.
Response: Thank you very much for your careful checks and suggestions. We are very sorry for our incorrect writing. According to your suggestion, the words “nuclear factor 191 kappa B” have been corrected to “Nuclear factor kappa-light-chain-enhancer of activated B cells.” in the revised manuscript. Please refer to the revised manuscript in the “Table 1”.
- It is also necessary to research other mycotoxins like mycophenolic acid and patulin,
"Huang, X., Huang, Q., He, Y., Chen, S. and Li, T., 2019. Mycophenolic acid enhanced lipopolysaccharide-induced interleukin-18 release in THP-1 cells via activation of the NLRP3 inflammasome. Immunopharmacology and Immunotoxicology, 41(5), pp.521-526." "Chu, Q., Wang, S., Jiang, L., Jiao, Y., Sun, X., Li, J., Yang, L., Hou, Y., Wang, N., Yao, X. and Liu, X., 2021. Patulin induces pyroptosis through the autophagic-inflammasomal pathway in liver. Food and chemical toxicology, 147, p.111867.
Response: Thank you very much for your professional suggestion. We have added the some sentences “(2) Most of the research on NLRP3 inflammasome activation focused on DON, AFB1, and ZEA, whereas studies of OTA, T-2 toxin, FB1, mycophenolic acid, and patulin are scarce; Alternaria and HT-2 toxins, citrinin, enniatins, ergot alkaloids, and nivalenol have not been studied at all. It is currently not possible to ascribe a set of common fac-tors to the activation of NLRP3 inflammasomes triggered by all 500 mycotoxins based on the existing research. Therefore, a wide range of mycotoxins as a subject of future study is necessary to improve understanding of the general rules for mycotoxin-induced NLRP3 inflammasome activation.” in the revised manuscript. Please refer to the revised manuscript in the “5. Conclusions and future perspectives” section. (The fourth paragraph).
Reviewer 2 Report
Comments and Suggestions for Authors
This is an excellent review that goes on to evaluate how the NLRP3 inflammasome in mycotoxin-induced toxicological mechanisms. This review is well written and the content is very interesting. The English is adequate. I think it can be published by responding to these minor requests:
1. Image quality needs to be improved. Some of these appear to be out of focus. Try, if you have not already done so, to save them in TIFF quality.
2. Re-check the entire manuscript to correct any typos.
3. 157: You should insert a space between “post” and “translational”.
4. 725 Make the same for “toxicityand”
5. Add more references riegarding work in recent years (last five years).
6. You should describe in more detail the pathways you mentioned and how interleukins are recruited.
Author Response
This is an excellent review that goes on to evaluate how the NLRP3 inflammasome in mycotoxin-induced toxicological mechanisms. This review is well written and the content is very interesting. The English is adequate. I think it can be published by responding to these minor requests:
Response: Thank you very much for your evaluation of the paper. We are very grateful for your positive and constructive comments and suggestions. Those suggestions and comments are all valuable and very helpful for revising and improving our paper. We tried our best to improve the quality of the manuscript. Some mistakes and errors, including but not limited to your suggestions, were modified in the revised manuscript.
- Image quality needs to be improved. Some of these appear to be out of focus. Try, if you have not already done so, to save them in TIFF quality.
Response: Thank you very much for your careful checks and suggestions. According to your suggestion, we have redrawn the image (Figure 1) and provided it in TIFF format.
- Re-check the entire manuscript to correct any typos.
Response: Thank you very much for your suggestion. According to your suggestion, we have rechecked the entire manuscript to eliminate any incorrect words.
- 157: You should insert a space between “post” and “translational”.
Response: Thank you very much for your careful checks and suggestions. According to your suggestion, we have added the punctuation “-” between “post” and “translational” in the revised manuscript. Please refer to the revised manuscript in the “3. Mycotoxins induce activation of the NLRP3 inflammasome” section (The second paragraph).
- 725 Make the same for “toxicityand”
Response: Thank you very much for your careful checks and suggestions. We are very sorry for our incorrect writing. According to your suggestion, the words “toxicityand” have been corrected to “toxicity and” in the revised manuscript. Please refer to the revised manuscript in the “5. Conclusions and future perspectives” section (The last paragraph).
- Add more references riegarding work in recent years (last five years).
Response: Thank you very much for your professional suggestion. We have added 13 references (ref. 8, ref. 17, ref. 18, ref. 40, ref. 41, ref. 42, ref. 43, ref. 71, ref. 72, ref. 97, ref. 102, ref. 104, and ref. 105) from the past five years in the revised manuscript. Please refer to the revised manuscript in the “References” section.
- You should describe in more detail the pathways you mentioned and how interleukins are recruited.
Response: Thank you very much for your professional suggestion. According to the published literature, the signaling pathway of mycotoxin-activated NLRP3 has not been demonstrated in more detail. According to your suggestion, we have added a description of the signaling pathway using NLRP3 inflammasome activation by DON as an example in the revised manuscript. “In the case of DON exposure, the first signal comes from TLR4 and NF‑κB and increases the in-tracellular transcripts of pro‑IL‑1β, pro‑IL‑18, and NLRP3. The NLRP3 inflammasome is activated by NLRP3 oligomerization with ASC and pro‑caspase‑1. Caspase-1 cleavesage of GSDMD forms active N-termini, which causes cell membrane perforation. Active caspase-1 then converts pro-IL-1β and pro-IL-18 to their biologically active structures, causing the release of IL-1β and IL-18 into the extracellular matrix, amplifying inflammation, and inducing pyroptosis. ” Please refer to the revised manuscript in the “Figure 1”.
Reviewer 3 Report
Comments and Suggestions for Authors
1. #1. Introduction. In the second paragraph, references should be added to the sentence ‘an increasing number of studies … related to pyroptosis’.
2. #1. Introduction. In the second paragraph, full name of ‘NLRC4, AIM2’ should be written when they first appear. Check the full-text.
3. #2. Mycotoxicosis. In the first paragraph, this sentence ‘Some mycotoxins have acute toxicity.’ is suggested to be deleted or integrated into other sentences.
4. #2. Mycotoxicosis. In the second paragraph, references should be added to the sentence ‘An increasing number of studies… in mycotoxin-induced toxicity’.
5. #2. Mycotoxicosis. In the second paragraph, the phase ‘At the in vitro level’ is advised for “in vitro”.
6. #3. Mycotoxins induce activation of the NLRP3 inflammasome. The contamination of animal feed or feed raw materials by mycotoxins should be described in section of ‘3.1., 3.2.,…’.
7. #Figure 1. There are two different forms of question marks in the picture. What's the difference?
8. # 3.3. ZEA and the NLRP3 inflammasome. In the second paragraph, references should be added to the sentence ‘Studies have shown that … interfering with the TLR4/NF-κB pathway’.
9. # 3.5. OTA and the NLRP3 inflammasome. In the third paragraph, references should be added to the sentence ‘studies have reported … in various cell lines’.
10. # 4. Targeting the NLRP3 inflammasome for mycotoxin exposure. What dose ‘for r against’ mean the last sentence in the first paragraph.?
11. # 4. Targeting the NLRP3 inflammasome for mycotoxin exposure. Dose concentration of the drugs (substances) should be mentioned when describing how inhibition of activation of the NLRP3 inflammasome effectively reduces the toxic effects of mycotoxins, e.g., ‘MCC950 has also been shown to attenuate…’.
12. # Table 1. What do the arrows in the table represent?
13. # Table 1. The font of the phase ‘Nrf2, N uclear factor erythroid 2 -related factor 2’ is false.
Author Response
Response: Thank you very much for your evaluation of the paper. We are very grateful for your positive and constructive comments and suggestions. Those suggestions and comments are all valuable and very helpful for revising and improving our paper. We tried our best to improve the quality of the manuscript. Some mistakes and errors, including but not limited to your suggestions, were modified in the revised manuscript.
- #1. Introduction. In the second paragraph, references should be added to the sentence ‘an increasing number of studies … related to pyroptosis’.
Response: Thank you very much for your professional suggestion. We have added the reference in the revised manuscript. Please refer to the revised manuscript in the “1. Introduction” section (The second paragraph).
- #1. Introduction. In the second paragraph, full name of ‘NLRC4, AIM2’ should be written when they first appear. Check the full-text.
Response: Thank you very much for your professional suggestion. We have added the full name of the abbreviations ‘NLRC4, AIM2, IFI16, HIN, IL-1β, DNA, RNA, ASC, NF-κB, ROS, TLR4, IPEC-J2, ETEC, PEDV, COX-2, mTOR, ’ when they first appear in the revised manuscript.
- #2. Mycotoxicosis. In the first paragraph, this sentence ‘Some mycotoxins have acute toxicity.’ is suggested to be deleted or integrated into other sentences.
Response: Thank you very much for your careful checks and suggestions. We have deleted the sentence ‘Some mycotoxins have acute toxicity.’ in the revised manuscript. Please refer to the revised manuscript in the “2. Mycotoxicosis” section (The second paragraph).
- #2. Mycotoxicosis. In the second paragraph, references should be added to the sentence ‘An increasing number of studies… in mycotoxin-induced toxicity’.
Response: Thank you very much for your professional suggestion. We have added the reference in the revised manuscript. Please refer to the revised manuscript in the “2. Mycotoxicosis” section (The second paragraph).
- #2. Mycotoxicosis. In the second paragraph, the phase ‘At the in vitro level’ is advised for “in vitro”.
Response: Thank you very much for your careful checks and suggestions. We are very sorry for our incorrect writing. According to your suggestion, the words “At the in vitro level” have been corrected to “in vitro” in the revised manuscript. Please refer to the revised manuscript in the “2. Mycotoxicosis” section (The second paragraph).
- #3. Mycotoxins induce activation of the NLRP3 inflammasome. The contamination of animal feed or feed raw materials by mycotoxins should be described in section of ‘3.1., 3.2.,…’.
Response: Thank you very much for your professional suggestion. We have added the information about the contamination of animal feed or feed raw materials by mycotoxins in the revised manuscript. Please refer to the revised manuscript in the “3.1. DON and the NLRP3 Inflammasome, 3.2. AFB1 and the NLRP3 inflammasome, 3.3. ZEA and the NLRP3 inflammasome, 3.4. T-2 toxin and the NLRP3 inflammasome, 3.5. OTA and the NLRP3 inflammasome, and 3.6. FB1 and the NLRP3 inflammasome” section (The third paragraph).
- #Figure 1. There are two different forms of question marks in the picture. What's the difference?
Response: Thank you very much for your careful checks and suggestions. The question mark represents status unknown. To make it easy to understand, we have deleted the two question marks. Please refer to the revised manuscript in the “Figure 1”.
- # 3.3. ZEA and the NLRP3 inflammasome. In the second paragraph, references should be added to the sentence ‘Studies have shown that … interfering with the TLR4/NF-κB pathway’.
Response: Thank you very much for your professional suggestion. We have added the reference in the revised manuscript. Please refer to the revised manuscript in the “3.3. ZEA and the NLRP3 inflammasome” section (The second paragraph).
- # 3.5. OTA and the NLRP3 inflammasome. In the third paragraph, references should be added to the sentence ‘studies have reported … in various cell lines’.
Response: Thank you very much for your professional suggestion. We have added the reference in the revised manuscript. Please refer to the revised manuscript in the “3.5. OTA and the NLRP3 inflammasome” section (The third paragraph).
- # 4. Targeting the NLRP3 inflammasome for mycotoxin exposure. What dose ‘for r against’ mean the last sentence in the first paragraph.?
Response: Thank you very much for your careful checks and suggestions. We are very sorry for our incorrect writing. According to your suggestion, the words “for r against” have been corrected to “for against” in the revised manuscript. Please refer to the revised manuscript in the “4. Targeting the NLRP3 inflammasome for mycotoxin exposure” section (The first paragraph).
- # 4. Targeting the NLRP3 inflammasome for mycotoxin exposure. Dose concentration of the drugs (substances) should be mentioned when describing how inhibition of activation of the NLRP3 inflammasome effectively reduces the toxic effects of mycotoxins, e.g., ‘MCC950 has also been shown to attenuate…’.
Response: Thank you very much for your professional suggestion. We have added the concentration of the drugs (substances) in the revised manuscript. Please refer to the revised manuscript ‘MCC950, BAY 11-7082, curcumin, luteolin, Lico A, rapamycin, 3-methyladenine, Bacillus licheniformis, dimethyl fumarate, and taurine’ in the “4. Targeting the NLRP3 inflammasome for mycotoxin exposure” section.
- # Table 1. What do the arrows in the table represent?
Response: Thank you very much for your careful checks and suggestions. According to your suggestion, we have added the words ‘↑ represents upregulated levels; ↓ represents downregulated levels’. in the revised manuscript. Please refer to the revised manuscript in the “Table 1”.
- # Table 1. The font of the phase ‘Nrf2, N uclear factor erythroid 2 -related factor 2’ is false.
Response: Thank you very much for your careful checks and suggestions. We are very sorry for our incorrect writing. According to your suggestion, the words “Nrf2, N uclear factor erythroid 2 -related factor 2” have been corrected to “Nrf2, Nuclear factor erythroid 2 -related factor 2” in the revised manuscript. Please refer to the revised manuscript in the “Table 1”.